# Illness Perceptions and Self-Management among People with Chronic Lung Disease and Healthcare Professionals: A Mixed-Method Study Identifying the Local Context

**DOI:** 10.3390/healthcare10091657

**Published:** 2022-08-30

**Authors:** Xiaoyue Song, Cynthia Hallensleben, Bo Li, Weihong Zhang, Zongliang Jiang, Hongxia Shen, Robbert J. J. Gobbens, Niels H. Chavannes, Anke Versluis

**Affiliations:** 1Department of Public Health and Primary Care (PHEG), Leiden University Medical Center, 2300 RC Leiden, The Netherlands; 2National eHealth Living Lab (NeLL), 2300 RC Leiden, The Netherlands; 3Faculty of Nursing and Health, Henan University, Kaifeng 475000, China; 4Faculty of Nursing and Health, Zhengzhou University, Zhengzhou 450001, China; 5Faculty of Health, Sports and Social Work, Inholland University of Applied Sciences, 1081 HV Amsterdam, The Netherlands; 6Zonnehuisgroep Amstelland, 1186 AA Amstelveen, The Netherlands; 7Department Family Medicine and Population Health, Faculty of Medicine and Health Sciences, University of Antwerp, 2610 Antwerp, Belgium

**Keywords:** illness perception, self-management, chronic lung diseases, local context, mixed-method study, patients, healthcare professional, primary care, secondary care, nursing

## Abstract

Self-management interventions (SMIs) may fail if they misalign with the local context. To optimize the implementation of SMIs in Chinese people with chronic lung disease (CLD), the local context was identified in Chinese primary care (PC) and secondary care (SC). A mixed-method study using semi-structured interviews and quantitative surveys was conducted on people with CLD and healthcare professionals (HCPs). The qualitative data was collected until data saturation was reached, and participants were invited to complete the survey after the interview. The qualitative data—analyzed with the framework approach—was triangulated with the quantitative data. A total of 52 participants completed the interviews, and 48 also finished the survey. Four themes were identified; (a) illness perceptions (e.g., patients had poor CLD knowledge and SM, inadequate resources lead to suboptimal disease control in PC); (b) self-management skills (e.g., most patients delayed exacerbation recognition and action, and some were admitted at the crisis point); (c) factors influencing self-management skills (e.g., (in)adequate disease knowledge and medical expenditure affordability); and (d) needs for self-management (e.g., increased disease knowledge, individualized self-management plan, eHealth, (healthcare insurance) policy support). Identified themes were dependent on each other and should be leveraged when implementing SMIs. Ultimately, such SMIs can optimize patient health outcomes.

## 1. Introduction

There is a high prevalence of CLDs in low- and middle-income countries, such as China [1,2]. Specifically, more than one-fourth of CLD patients are in China; over 144 million Chinese people are affected by CLDs [1,2]. The high disease burden for CLD (mainly chronic obstructive pulmonary disease [COPD] and asthma) is due to ineffective intervention [3,4].

There is evidence that (blended) self-management interventions (SMIs) could significantly improve patients’ quality of life and reduce emergency department visits [5]. Self-management (SM) is defined as an individual’s ability to manage symptoms, treatment, physical and psychosocial consequences, and lifestyle changes inherent to life with a chronic illness [6]. In China, CLD SM is suboptimal [2,3,7]. Specifically, people with CLD in China take late action resulting in exacerbations, which indicates the need for effective SMIs [8]. Exacerbations are defined as sustained worsening of a patient’s condition beyond normal day-to-day variations that are acute in onset, which may also require a change in medication with or without hospitalization [7].

Implementing effective SMIs in China may help to reduce the disease burden in people with CLD [5,9]. Many factors need to be considered when implementing SMI; the critical factor is the compatibility between SMI and the local context [10,11]. Local context is defined as the set of characteristics and circumstances surrounding the implementation effort, such as the local beliefs, local health behaviours, and socioeconomic aspects [12]. Identifying the local context, e.g., local illness perceptions and experience with and needs for SM, is essential to facilitate the alignment between SMI implementation and local context [10,11].

Illness perceptions involve the illness’s identity, causes, consequences, length (timeline), and whether it can be cured or controlled [13,14]. Determining what illness symptoms are present can help identify how patients cope with or self-manage the disease [15]. This study aims to gain insight into two SM skills, i.e., exacerbation recognition and action. Both skills are essential because they help to reduce recovery time and decrease disease burden [9].

The current study aims to map the local context of CLD in China. People with CLD and healthcare professionals (HCPs) in Chinese primary care (PC) and secondary care (SC) will be included. HCPs are included because of their essential role in helping patients manage their diseases. Since patients and HCPs may hold different views in PC and SC, this study will be conducted in both settings. Altogether, we aimed to identify the local context, including illness perceptions, experience with, and needs for SM in people with CLD and HCPs in Chinese PC and SC.

## 2. Materials and Methods

### 2.1. Design

A mixed-method study involving semi-structured interviews and a survey was used [16].

### 2.2. Settings and Participants

The study was conducted in people with CLD and HCPs working with CLD—from November 2019 to May 2020—in PC and SC in Zhengzhou and Kaifeng. The inclusion criteria of patients were: (1) ≥18 years old, (2) diagnosed with CLD or repeated persistent cough lasting longer than eight weeks in the past two years [17], and (3) fluency in spoken and written Chinese. People with mental disabilities, as diagnosed by the physician, were excluded. HCPs were included when they worked in PC or the respiratory department of SC. Recruitment of participants was carried out through random and purposive techniques [18].

### 2.3. Measurements and Outcomes

#### 2.3.1. Qualitative Interview

An interview topic list focused on illness perceptions towards CLD, experience with, and the needs for SM. Two vignettes—one focusing on COPD and one on asthma—introduced CLD to patients and HCPs. Both vignettes were checked by the CLD nurse specialist (CH) to ensure their validity and feasibility before being used. Interviews lasted between 45 and 70 min. The interview topic list can be found in Appendix A.

#### 2.3.2. Quantitative Survey

In patients, demographic and clinical characteristics were collected: age, gender, years with disease or symptoms, and the exacerbation frequency in the last year. Illness perceptions were measured with the 8-item Brief Illness Perceptions Questionnaire (BIPQ) [19]. The BIPQ helps identify patients’ opinions on their disease: identification (symptoms experienced), illness coherence (understanding of disease), consequences, emotional responses, illness concern, timeline, and personal and treatment control. Each item is scored from 0–10, with higher scores indicating a more threatening view of the illness [19]. Items on personal control, treatment control, and emotional response were reverse scored. The total score on the BIPQ ranged from 0 to 80. The items and their implications can be found in Table 1. The BIPQ has good internal reliability and has been used with various illness groups [19]; the Chinese version of BIPQ has acceptable test-retest reliability, with a Cronbach’s alpha of 0.54 to 0.76 [20]. Participants had to select the three most important causes of their illness from 18 possible causes. Next, there were questions on smoking behaviours used in other studies [21,22]. Ex-smokers were asked about the number of years they had smoked. Current smokers were asked the following eight questions: the number of years they smoked, the number of cigarettes smoked daily, type of smoking products, opinion on smoking damage, history of trying to stop smoking, the longest period managed to stop smoking, and interests and confidence to stop smoking.

For HCPs, the demographic characteristics included gender and years of work. An adjusted version of the BIPQ was used to identify HCPs’ perception of the patients’ disease with the mentioned eight illness representations [19]. Next, HCPs had to select the three most important causes of illness from the exact 18 causes shown to patients. Furthermore, HCPs’ perceptions of CLD guideline recommendations and confidence in implementing guideline recommendations were assessed [23]. Respondents were asked to indicate their level of agreement using a five-point scale (frequency ranged from ‘never’ to ‘always’; confidence ranged from ‘not at all confident’ to ‘extremely confident’). The quantitative questionnaires for patients and HCP can be found in Appendix A.

### 2.4. Data Collection

Before the interview, the researcher (XYS) provided detailed study information and asked for written consent. After obtaining the written consent from the participants, the researcher (XYS) used the topic list to guide the interview. Interviews were audiotaped, and notes were made if necessary. Then, participants were asked to complete the quantitative surveys. The data were collected at the healthcare settings or the participant’s home.

### 2.5. Data Analysis

The framework approach [24] guided the qualitative analysis. Two researchers (XYS, ZLJ) transcribed and read all the interviews. In the first three transcripts, codes were made in the margin of the transcripts. Next, the two researchers discussed and agreed on 52 codes to apply in subsequent transcripts. Codes were defined and grouped into categories to form a working analytical framework. Emerging codes from subsequent interviews continuously improved the framework. The categories and codes were applied to index the interviews. A separate sheet, with one row (per interview) and one column (per code), was used for each category. The codes and the quotations (i.e., sentences indexed with the codes) from each interview were summarized for each category. Researchers systematically identified themes based on the study. Atlas. Ti 7.5 software and Excel were used to store and manage the qualitative data.

Quantitative data were entered by the researchers (XYS and ZLJ) and analyzed using the IBM SPSS software package version 23.0. Descriptive analyses (e.g., mean, standard deviation [SD], *N*, percentages) were used to summarize the quantitative data (e.g., demographic, clinical characteristics, BIPQ data). The mean BIPQ score was compared between groups using an independent *t-*test (i.e., patients from PC versus SC settings, HCPs from PC versus SC settings, and all patients versus all HCPs). All statistical tests were two-sided, with the significance at *p* ≤ 0.05.

### 2.6. Validity and Reliability

XYS conducted the first interview, and three researchers checked the transcripts (CH, AV and RG) to ensure sufficient interview quality. Furthermore, two interviewed HCPs read their transcript to ascertain that the interviewer represented their perspectives accurately. These two participants did not suggest any changes. The self-developed questions, including smoking behaviors, perceptions of guideline recommendations, and confidence in implementing guideline recommendations, were based on the previously applied questionnaire with proven feasibility and acceptability.

Two Chinese researchers independently coded all interview transcripts (XYS and ZLJ). The Chinese researcher (XYS) translated the initial English analytical framework and checked it by the XYS and English-speaking researchers (CH and AV) to ensure validity. In the quantitative data analysis, two researchers independently entered the data into the SPSS software (XYS and ZLJ) to ensure data consistency after checking.

## 3. Results

### 3.1. Descriptive Statistics

A total of 27 patients and 25 HCPs participated in the interviews; 25 patients and 23 HCPs completed the quantitative survey. Reasons for not completing the survey were: having a health check-up or treatment (patients) and having an emergency meeting or insufficient time (HCPs). Detailed descriptive statistics on patients and HCPs are in Table 2 and Table 3.

### 3.2. Theme 1: Illness Perception 

The qualitative data on illness perceptions were categorized using subthemes; see headings below. The quantitative data on illness perceptions is shown in Table 4.

#### 3.2.1. Theme 1a: Coherence and Identification

Most patients in PC and all patients in SC reported that the last exacerbation was a difficult time for them. That is because they suffered physical and psychosocial function deterioration. Yet two patients in PC felt the last exacerbation was not a problem because they experienced regular exacerbations during the winters. Patients in different healthcare settings described different symptoms, i.e., patients in PC experienced coughing, wheezing, and chest tightness while those in SC underwent dyspnea (Table 5, Q1).

After reading vignettes on COPD and asthma, patients and HCPs identified the diseases differently (see Figure 1). Moreover, HCPs in PC diagnosed diseases by their working experience and those in SC using spirometry equipment (Table 5, Q2, Q3).

#### 3.2.2. Theme1b: Perceived Causes

When patients discussed the disease’s cause, most attributed it to age, air pollution, and smoking. Because the Chinese term ‘cause’ can also mean ‘to provoke’, food and physical activity were mentioned as cause/triggers (Table 5, Q4, Q5). HCPs noted that the air pollution, age, smoking and seasonal changes from fall to winter contributed to the exacerbations (Table 5, Q6). Quantitative data showed that patients and HCPs perceived air pollution, smoking, and age as the prevailing disease causes, and patients additionally perceived weather as the disease cause. The distribution of the perceived causes are in Appendix A.

Some participants identified the vignette as chronic lung disease without specific names; some patients reported that it was not their obligation to recognize a disease name unfamiliar to them.

#### 3.2.3. Theme 1c: Perceived Consequences and Emotional Response

Patients mentioned that the exacerbations gradually deteriorated health-related quality of life. Some 15 patients’ physical function was affected by the symptoms, and 5 of them experienced a negative impact on their sleep quality. The lack of a restful night’s sleep and the morning battle of coughing and mucus expulsion left patients feeling exhausted, and the symptoms affected their mood the following day. Two patients said their daily life was limited. A total of 23 patients experienced reduced social interaction (Table 5, Q7, Q8). The misunderstanding from other people, for example, that the symptoms are contagious, pushed the patients away from social activities. Patients also frequently felt guilty due to their productivity losses (Table 5, Q9).

HCPs reported that physical limitation and decreased lung function were significant consequences of the diseases (Table 5, Q10, Q11). Moreover, most HCPs mentioned not paying enough attention to patient complaints (Table 5, Q12).

#### 3.2.4. Theme 1d: Curable Possibility and Perceived Duration

A total of 20 patients believed that the disease was incurable and chronic after being informed by their HCPs, while 7 patients in SC stated that their disease was curable and acute, and they were cured when asymptomatic (Table 5, Q13). All HCPs highlighted that CLD was incurable and chronic (Table 5, Q14, Q15).

#### 3.2.5. Theme1e: Identified Disease Control

Patients believed they were powerless to control the disease, while HCPs were sufficiently professional to help them manage it (Table 5, Q16, Q17). Therefore, they commonly perceived that they would SM the disease well when following medical advice. HCPs admitted that SM was helpful in managing diseases, but their patients showed poor SM (Table 5, Q18). Additionally, HCPs in PC mentioned that limited CLD medications were available in their settings. Moreover, all HCPs were encouraged to adopt the CLD guidelines in practice. Despite broad familiarity with the guidelines in SC, knowledge about guidelines in PC was suboptimal. In addition, HCPs did not always adhere to CLD guidelines for different reasons (Table 5, Q19, Q20). Moreover, most HCPs in PC sometimes or never applied the guideline recommendations, while those in SC always or often used the recommendations (see Figure 2).

### 3.3. Theme 2: Identified SM Skills

Most patients struggled to use SM skills, i.e., exacerbation recognition and action, while some patients recognized and took action on exacerbations early (Table 5, Q21–24). The distribution of identified SM skills in patients between PC and SC can be found in Appendix A. Most HCPs noted that patients postponed exacerbation actions, yet two HCPs from PC observed that some patients went to them early (Table 5, Q25–27).

### 3.4. Theme 3: Factors Influencing SM Skills

Three generic factors were identified to influence exacerbation recognition and actions, including disease knowledge, former experience with exacerbations, and family support. Moreover, one specific factor influencing the ability to recognize the exacerbations was identified (i.e., perceived illness severity), and three specific factors influencing the performance of SM skills were identified (i.e., self-empowerment, Chinese herb, and medical expenditure affordability). These factors—with the facilitating and barrier aspects—were dependent on each other; details on how these factors influence SM skills can be found in Table 6, Q28–41.

### 3.5. Theme 4: Needs for SM

The needs for SM addressed the expected facilitators, e.g., increased disease knowledge and the strategy to support the SM, e.g., eHealth use and individualized SM plan. The other details on the needs for SM are included in Table 7, Q42–51.

## 4. Discussion

This mixed-method study identified the local context of people with CLD and HCPs in Chinese PC and SC. Four themes were identified, namely, (a) illness perceptions; (b) identified SM skills; (c) factors influencing SM skills; and (d) needs for SM. These themes were dependent on each other and should be addressed when implementing SMIs.

Most patients could not identify the CLD correctly; this finding was evidenced by a previous study showing that patients had limited disease knowledge [25]. Moreover, the disease decreased patients’ physical and psychosocial functioning. Most patients believed the disease was chronic, yet a few believed it would not be long-term, possibly due to their limited understanding. Age, air pollution, and smoke were the leading disease causes in interviews, and the survey identified the weather as an additional cause. The weather was mentioned as the temperature variability can trigger exacerbations [26].

Compared with those in PC, patients in SC showed a more severe emotional response to the illness. This finding is not surprising because patients in SC suffer more severe exacerbations, which leads to more negative emotions [27]. Notably, no difference was found in total illness perception scores from patients between PC and SC, which could be due to the chronic nature of CLD. Most patients in this study had CLD for more than five years and went to PC and SC for mild and severe exacerbations; therefore, patients with long-term CLD may hold similar illness perceptions. Interestingly, no other existing studies have compared illness perceptions—in people with CLD—between PC and SC.

Most patients delayed exacerbation recognition and actions. It is likely that these patients had difficulties with exacerbation recognition and had limited knowledge of potential actions they could undertake. A few patients, mainly in PC, showed early exacerbation recognition and action. Their early response to the disease was because they applied their disease knowledge and former experience to identify the exacerbation and take prompt action. Early presenters were mostly from PC because of the accessibility, e.g., the location was closer to the patient’s home, and the treatment cost was lower [28]. The identified illness perceptions among people with CLD were in line with the previous study [25].

Most HCPs correctly identified the CLD. HCPs agreed that CLD was chronic and could affect physical and psychosocial functioning. This finding complies with one previous study [29]. HCPs agreed that patients showed poor SM skills.

In general, HCPs in PC—compared with HCPs in SC—held a more threatening view of the CLD. The difference was possibly related to the healthcare service setup in China. HCPs in SC are better equipped with medical skills and more patient experience, contributing to their positive attitudes [30]. Compared with patients, HCPs held a less threatening view of CLD. It could be explained that HCPs had more disease knowledge after their medical training, which led to a better understanding of how to control CLD.

Unlike patients diagnosed with a spirometry test in SC, CLD was diagnosed based on disease history and clinical symptoms in PC due to the lack of test facilities. All HCPs (strongly) agreed with but did not adhere to the guideline recommendations for different reasons; namely, HCPs in SC considered the symptoms of people with CLD more complicated than described in the guidelines, while in PC, there was limited guideline knowledge. The lack of confidence in disease treatment in PC could be explained by a lack of spirometry tests and lower professional knowledge [31].

Many generic factors were found to influence patient SM. The first factor was disease knowledge. Patients who had more disease knowledge showed a higher SM ability. The second was the former experience with exacerbations. Some patients showed prompt action due to unwillingness to suffer the adverse outcomes of the delayed exacerbation action. Nevertheless, with the former exacerbation experience, some patients took late SM actions. That is possibly because they were used to living with the symptoms, making it difficult to recognize the exacerbation and take prompt action. The third was family support; it helped patients self-manage exacerbations, yet family members expected them to do housework beyond their physical ability. A lack of disease knowledge among family members is a potential explanation [32].

One specific factor influencing exacerbation recognition was identified as the perceived illness severity. Different factors were found to influence SM actions on exacerbation. The first was patient self-empowerment. The second was Chinese herbs. Patients used such herbs to reduce the exacerbations, while HCPs perceived patients should take these herbs with their suggestions. The finding on the Chinese herbs was not reported in other studies. The worry from HCPs was likely because patients taking the Chinese herbs without professional guidance may decrease the SM on exacerbations. The third was the medical expenditure affordability; the higher affordability, the earlier action on the exacerbations. The identified third factor aligns with the other study [33].

The needs for SM reflected the strategies to improve SM and expected facilitators to optimize patient SM. Both patients and HCPs expressed the need for increased disease knowledge, which was the facilitator for the SM. Moreover, public healthcare insurance was needed to cover medical expenditures for the CLD. In addition, the need for an individualized SM plan, eHealth use, and independent CLD specialist reflected effective strategies to deliver SMIs easily accessible and not restricted to certain places and times. The identified needs correspond with the previous study [25]. Notably, the HCPs further pointed out that obtaining (healthcare insurance) policy support from the government was necessary. Such a finding is not surprising as optimal SMIs have been more easily adopted with support from the government in China [34].

Several limitations need to be addressed. First, less detailed information could be obtained from participants because, during the study, there was a coronavirus outbreak in China. Some participants did not elaborate much on their SM experiences because they feared the risk of becoming infected if they talked to the interviewer for too long. Second, four participants did not participate in the survey. The missing quantitative data may underpower the triangulation. Third, the small sample size in the quantitative survey may be underpowered to triangulate the qualitative data, which may negatively impact this study’s validity. Therefore, the results should be interpreted cautiously due to the mentioned limitations.

## 5. Conclusions

This study presented a comprehensive view of the local context—on the identified illness perceptions and experience with and needs for SM—in China. The identified findings addressed the importance of increasing disease knowledge, developing the strategies to deliver the SMI, and gaining (healthcare insurance) policy support during the SMI implementation. Ultimately, such SMIs can help to improve patient health outcomes and reduce the disease burden. Furthermore, this study provided that the local context should be emphasized and leveraged to ensure the SMI meets local needs. A large-scale quantitative study is needed to support the findings.

## Figures and Tables

**Figure 1 healthcare-10-01657-f001:**
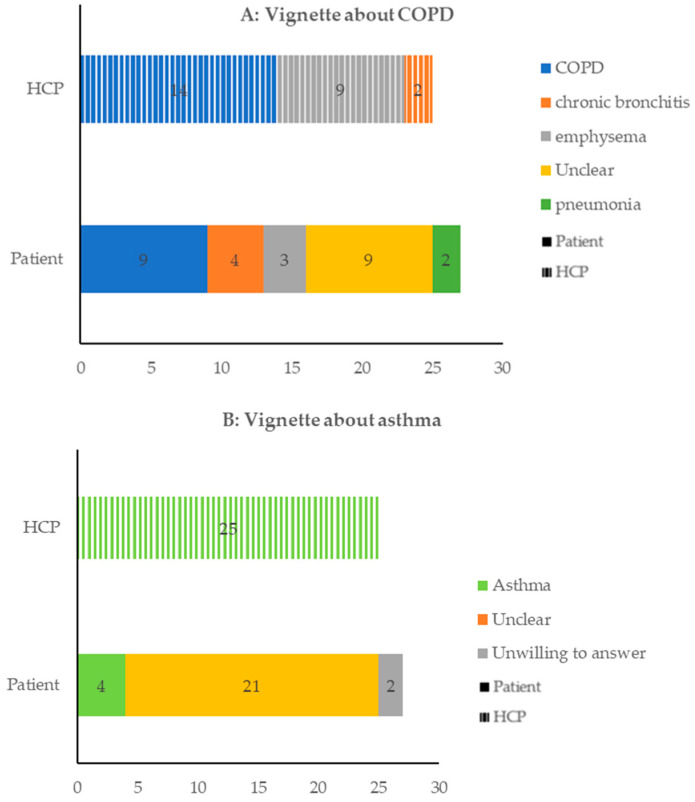
Identification of the chronic obstructive pulmonary disease (COPD) and asthma vignette from patients and healthcare professionals (HCPs).

**Figure 2 healthcare-10-01657-f002:**
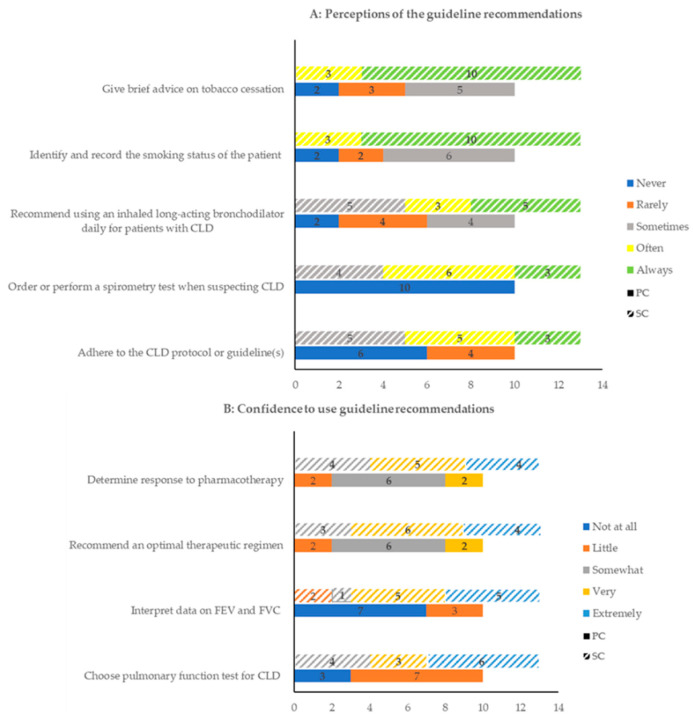
Perceptions of and confidence in using the guideline recommendations in practice by healthcare professionals (HCPs) in primary care (PC) and secondary care (SC).

**Table 1 healthcare-10-01657-t001:** Interpretation of the Brief Illness Perception Questionnaire.

Items	A Higher Score Implies:
Consequences	Greater perceived influence of the illness
Timeline	A stronger belief in a chronic time course
Personal control	Greater perceived personal control
Treatment control	Greater perceived control of the treatment
Identity	Greater experience of severe symptoms as a result of the illness
Concern	Greater feelings of concern about illness
Coherence	A better understanding of the illness
Emotion	A stronger emotional response to the illness

**Table 2 healthcare-10-01657-t002:** Descriptive statistics of patients and healthcare professionals (HCPs).

Data about Patients	*N*	Data about HCPs	*N*
Location		Location	
Primary care	14	Primary care	13
Secondary care	11	Secondary care	10
Disease diagnosis		Gender	
COPD	18	Male	4
Emphysema	3	Female	19
Asthma	2	Years of working experience	
Chronic bronchitis	2	<5	2
Age (years) (mean ± SD)	69.60 ± 13.07	5–10	19
Years with disease		≥10	2
<5	5		
5–10	7		
≥10	13		
Number of exacerbations in the last year			
0	2		
1	10		
≥2	13		
Smoking status			
Current smokers	5		
Ex-smokers	16		
Never smoked	4		
Mean years of smoking			
Current smokers	33.75 ± 14.24		
Ex-smokers	48.80 ± 13.07		

COPD: chronic obstructive pulmonary disease; SD: standard deviation.

**Table 3 healthcare-10-01657-t003:** Descriptive statistics of patients and healthcare professionals (HCPs).

Data about Patients	*N*
Current smokers’ cigarette situation	
Mean daily smoking (cigarettes)	9.40 ± 3.78
Duration of quitting smoking (months)	
<6	2
6–12	3
≥12	0
Frequency of quitting smoking	
<2	1
≥2	4
Interest in quitting smoking	
Not at all	1
A little	1
Somewhat	2
Much	0
Very much	1

COPD: chronic obstructive pulmonary disease; SD: standard deviation.

**Table 4 healthcare-10-01657-t004:** Comparison of Brief Illness Perception Questionnaire scores between patients and healthcare professionals (mean ± SD).

Domains	Patients (n = 25)			HCPs (n = 23)		
	PC (n = 14)	SC (n = 11)	Total (n = 25)	PC (n = 10)	SC (n = 13)	Total (n = 23)
Consequences	5.21 ± 0.80	5.91 ± 0.94	5.52 ± 0.92	5.90 ± 0.74 *^b^	4.46 ± 0.97	5.09 ± 1.12
Timeline	8.07 ± 2.53	9.18 ± 0.40	8.56 ± 1.96	9.40 ± 0.52	9.08 ± 0.49	9.22 ± 0.52
Personal control	5.29 ± 0.61	5.45 ± 0.69	5.36 ± 0.64	4.10 ± 0.88 *^b^	3.38 ± 0.51	3.70 ± 0.76 **^c^
Treatment control	6.71 ± 1.07	7.36 ± 1.21	7.00 ± 1.15	2.60 ± 0.52	3.08 ± 0.64	2.87 ± 0.63 **^c^
Coherence	6.64 ± 0.84	7.19 ± 0.75	6.88 ± 0.83	5.30 ± 1.42 *^b^	6.39 ± 1.04	5.91 ± 1.31 **^c^
Concern	7.27 ± 1.10	7.36 ± 0.93	7.32 ± 0.99	5.60 ± 1.17 **^b^	2.92 ± 0.28	4.09 ± 1.56 **^c^
Identity	6.27 ± 0.79	5.79 ± 0.80	6.00 ± 0.82	5.60 ± 0.52	5.77 ± 0.73	5.70 ± 0.63
Emotional response	7.45 ± 0.69	8.21 ± 0.80 *^a^	7.88 ± 0.83	4.54 ± 1.51 *^b^	3.30 ± 0.48	4.00 ± 1.31 **^c^
Total score	53.29 ± 3.99	56.09 ± 3.05	54.42 ± 3.81	41.80 ± 2.57 *^b^	39.62 ± 1.39	40.57 ± 2.23 **^c^

Notes: *a: significantly different compared with patients in PC, *p* < 0.05; *b: significantly different compared with HCPs in SC, *p* < 0.05; **b: significantly different compared with HCPs in SC, *p* < 0.001; **c: significantly different compared with patients, *p* < 0.001. Abbreviations: SD = standard deviation; HCP = healthcare professional; PC = primary care; SC = secondary care.

**Table 5 healthcare-10-01657-t005:** Themes presented from patients and healthcare professionals (HCPs) of illness perceptions and self-management (SM) skills with quotes.

Data From Patient			Data from Healthcare Professionals	
Quotes	Category	Theme	Category	Quotes
Q1: “It felt like a plastic bag on my face, and I could not breathe in the oxygen at that moment.” (Patient 15, SC).	Illness coherence	1a. Illness coherence and identification	Illness coherence	Q2: “It is easy to diagnose it from my experience.” (HCP1, PC)
				Q3: “We always recommend that patients have a spirometry test.” (HCP 14, SC)
Q4: “Eating fried sunflower triggered my episode.” (Patient 2, PC)	Illness disease	1b. perceived causes	Disease cause	Q6: “Air pollution is the most important reason.” (HCP 23, SC)
Q5: “When I moved the goods, I was out of breath and fainted.” (Patient 15, SC)				
Q7: “I used to participate in square dancing.” (Patient 4, PC) Q8: “When I cough in public, people cover their mouths with their hands and go away from me. Their actions depress me.” (Patient 19, SC)	Reduced social interaction	1c. perceived consequences and emotional response	Physical limitation	Q10: “Patients work less after the exacerbation due to decreased physical function.” (HCP1, PC)
			Decreased lung function	Q11: “A new exacerbation accounts for the decreased lung function.” (HCP10, PC)
Q9: “I can’t work now. I am like a burden to my family.” (Patient 16, SC)			Indifference on patient complains	Q12: “I do not have time to explore patients’ feelings.” (HCP6, SC)
Q13: “No episodes disturb my daily life. I am just as healthy as those who do not have COPD or other chronic diseases.” (Patient 17, SC)	Asymptomatic equal to cured	1d. curable possibility and perceived duration	Incurable and chronic	Q14: “Chronic lung disease will accompany the patients for a lifetime.” (HCP6, PC) Q15: “It could not be cured.” (HCP16, SC)
Q16: “I cannot manage the disease by myself.” (Patient 7, PC)	Poor self-management	1e. identified disease control	Poor patient self-management	Q18: “SM is helpful to control the disease, but few patients can make it due to limited disease knowledge.” (HCP17, SC)
Q17: “Doctors are the professionals. I do what they asked me to do.” (Patient 20, SC)	Passive role with doctors		Guideline use in practice	Q19: “Patients’ symptoms were more complicated than described in the guidelines.” (HCP19, SC) Q20: “Few of us know the guidelines.” (HCP9, PC)
Q21: “Well, when I had the early symptoms, I thought I had a cold.” (Patient 8, PC)	Late exacerbation recognition	2. identified SM skills	Patient delayed action	Q25: “Some patients do not visit us until their family members force them.” (HCP10, PC) Q26: “Patients did not contact us until they reached a crisis point leading to hospitalization.” Q27: “He sends a message or dials a voice call to me via Wechat when he feels uncomfortable.” (HCP16, SC)
Q23: “If my symptoms worsen, I will ask my daughter to contact my doctor immediately.” (Patient 13, PC) Q24: “Early action can reduce the risk of being sent to the hospital.” (Patient 14, PC)	Early exacerbation action		Patient prompt action	

Q: quotes; SC: secondary care; PC: primary care.

**Table 6 healthcare-10-01657-t006:** The theme presents factors influencing self-management (SM) skills with quotes.

Themes & Category	Explanation	Quotes
Generic factors influencing SM skills		
Disease knowledge	Sufficient knowledge facilitated patients to develop SM skills, while insufficient knowledge was the barrier.	Q28: “I know nothing about the disease or the episode so that I could do nothing about it.” (Patient 2, PC) Q29: “My knowledge of the disease helps a lot.” (Patient 13, PC)
Former experience with exacerbations	Realizing the importance of early detection and prompt action from past experiences were the facilitators. Habituation to the disease from the former experience was the barrier.	Q30: “After my previous painful experience in the hospital, I realized that attention to the different symptoms (exacerbation) and visiting doctors early was essential.” (Patient 15, SC) Q31: “…However, I frequently did nothing about the worsening symptoms because I learned to live with it.” (Patient 16, SC)
Specific factors influencing the exacerbation recognition		
Family support	Helpful family support was the facilitator. In contrast, insufficient family support when patients were at home was the barrier.	Q32: “My daughters sent me to the hospital and covered the diagnosis cost. Their actions are warm and helpful.” (Patient 14, PC) Q33: “Family members regarded patients as well-functioned labor and expected them to do higher intensity household chores than the patient could endure. Such patients went on with the housework with all exacerbations.” (HCP 23, SC)
Perceived illness severity	Perceiving the exacerbation as usual was a barrier. The perception that the exacerbation was a hazardous event facilitated recognizing exacerbations early.	Q34: “I must pay attention to my disease carefully. Otherwise, I will be published by the worsened exacerbations.” (Patient 23, SC) Q35: “Some patients would not think breathlessness or coughing was a problem unless these symptoms disturb their eating and drinking.” (HCP 23, SC)
Self-empowerment	High self-empowerment facilitated the patients’ act on the exacerbations and vice versa.	Q36: “For the early symptoms, I can control them myself effectively. I will contact my daughter for the ambulance for symptoms out of my control.” (Patient 13, PC) Q37: “I always try to avoid the medicine or the doctors, even if I know my symptoms get worse.” (Patient 14, PC)
Chinese herb	Patients perceived the Chinese herb as facilitators, while HCPs perceived patients should take these medications with their suggestions.	Q38: “Chinese herb relieved aggravation.” (Patient 4, PC) Q39: “Patients take the Chinese herbs by themselves without informing us. To make sure the medicine works well, they should ask our advice before taking unprescribed Chinese medicine from us.” (Patient 5, PC)
Medical ex-penditure af-fordability	The higher medical expenditure affordability, the more likely the patient is to see a doctor early.	Q40: “Visiting the doctors means paying money, which is the last thing I want to do.” (Patient 4, PC) Q41: “My retirement pension and medical insurance can cover all the medical costs. When I am uncomfortable, I just visit the doctors.” (Patient 12, PC)

Q: quotes; PC: primary care; HCP: healthcare professional; SC: secondary care.

**Table 7 healthcare-10-01657-t007:** The theme presents needs for self-management (SM) with quotes.

Categories	Explanation	Quotes
Increased disease knowledge		Q42: “The doctor told me I was diagnosed with COPD and left…I expected to know more about this disease.” (Patient 15, SC) Q43: “With more information on disease and medications, patients will be more familiar with the seriousness of their condition, manage risk factors and change behavior, and then take action to meet their own needs for disease management.” (HCP 23, SC)
Individualized SM plan		Q44: “I need one intervention to help me recognize the episode early and act on it early.” (Patient 10, PC)
Nurse specialist		Q45: “A nurse specialist experienced in dealing with patients will be helpful to deliver SM information.” (HCP 17, SC)
eHealth use		Q46: “eHealth will help us deliver SM information to patients, e.g., Wechat.” (HCP 21, SC)
		Q47: “If we could remotely monitor patients’ diseases, we could provide more care to more patients.” (HCP 21, SC)
Sufficient family support		Q48: “When I forgot to take medicine, my family members remaindered me about it immediately. Family support can help me a lot.” (Patient 10, PC)
Policy support		Q49: “With the economic policy support from the government, we will have more resources to provide SM.” (HCP 2, PC)
		Q50: “If public medical insurance can cover more medical costs, more patients will choose to visit the doctors earlier.” (HCP 22, SC)

Q: quotes; PC: primary care; HCP: healthcare professional; SC: secondary care.

## Data Availability

The datasets generated and analyzed during the current study are available from the corresponding author on reasonable request.

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
