# Peer review of "Illness Perceptions and Self-Management among People with Chronic Lung Disease and Healthcare Professionals: A Mixed-Method Study Identifying the Local Context"

_healthcare, 2022, doi:10.3390/healthcare10091657_

Round 1

Reviewer 1 Report

The authors show that self-management interventions (SMIs) can help improve patient health outcomes and reduce disease burden, and discrepancies between PC and SC environments should be considered before interventions are performed in a healthcare setting. What the authors are trying to say is briefly summarized, but the overall flow or length of the article is too long.

1.     First, the introduction is too long. Summarize and shorten the important story. The aim of this study is to identify regional contexts that include disease awareness, experience, and needs for SM in people with CLD and HCP. Write a concise introduction to this content.

2.     Why is " repeated persistent cough lasting longer than eight weeks in the past two years" included in the including criteria? How many patients are there ?

3.     Table 3 shows the overall differences between PCs and SCs (both patients and HCPs). Describe the authors' opinions on the occurrence of these differences in the discussion.

Author Response

Response to Reviewer 1 Comments

Point 1: The authors show that self-management interventions (SMIs) can help improve patient health outcomes and reduce disease burden, and discrepancies between PC and SC environments should be considered before interventions are performed in a healthcare setting. What the authors are trying to say is briefly summarized, but the overall flow or length of the article is too long.

Response 1: Thanks for the useful comments. In line with the suggestion, we have shortened the article’s length by describing it more to the point and with less repetition. Specifically, the Introduction (please see p.1, line 44 to p.2, line 76) and Discussion (please see p.12, line 333 to p.14, line 410) sections were shortened to provide a more concise story. The Results section was rewritten to present the summarized findings (p.4, line 173 to p.12, line 327). We hope that the length of the article has hereby been improved.

Point 2: First, the introduction is too long. Summarize and shorten the important story. The aim of this study is to identify regional contexts that include disease awareness, experience, and needs for SM in people with CLD and HCP. Write a concise introduction to this content.

Response 2: In line with the suggestion, the Introduction has been shortened (please see p.1, line 44 to p.2, line 76).

Point 3: Why is " repeated persistent cough lasting longer than eight weeks in the past two years" included in the including criteria? How many patients are there?

Response 3: Individuals with a "repeated persistent cough lasting longer than eight weeks in the past two years" were also included because these symptoms are the novel phenotype of chronic obstructive pulmonary disease [18]. We have added the reference on this topic to clarify why we included these individuals (please see p.2, line 87). There were three people with persistent coughing; however, these people also had CLD diagnosis. No participant with only showed a persistent cough was thus reported in the Results section.

Point 4: Table 3 shows the overall differences between PCs and SCs (both patients and HCPs). Describe the authors' opinions on the occurrence of these differences in the discussion.

Response 4: Thank you for the suggestion. In line with your suggestion, we now discuss the overall differences between PC and SC patients (please see p.12, lines 341-348), between PC and SC HCPs (please see p.13, lines 360-363), and between patients and HCPs (please see p.13, lines 363-365).

Reviewer 2 Report

The sample is too small, and the qualitative analysis does not bring clear conclusions. A graphical presentation of the quantitative results would be valuable. There are many repeated phrases in the text.

Author Response

Point 1: The sample is too small, and the qualitative analysis does not bring clear conclusions. A graphical presentation of the quantitative results would be valuable. There are many repeated phrases in the text.

Response 1: We would like to thank you for your feedback. Regarding the sample size, we used the same sample size (and the same sample) in the qualitative and quantitative part, which is in line with mixed-method study designs [17]. The quantitative data was mainly used to triangulate the qualitative data, and the sample size for the qualitative data is in line with qualitative research guidelines; qualitative data was collected until data saturation was reached [19]. Nevertheless, we agree that the sample size in the quantitative part is small and have mentioned this in the Discussion as a limitation (please see p.14, lines 407-409).

We have made several changes to the manuscript to clarify the conclusions of this study. We rephrased the Conclusion section (please see p.14, lines 412-419) and the result and conclusion of the Abstract (please see p1, lines 24-39).  The Results section was rewritten to present the summarized findings (p.4, line 173 to p.12, line 327). Additionally, four figures, two in the main text and two in Appendix B, were added to graphically present the quantitative results (please see Figures 1-2 and Appendix B).

Furthermore, we have tried to improve the manuscript by deleting repetitions and writing more concisely. For example, the repeated phrases in the Result section were deleted, and the findings were put in two figures and Table 2 (please see p.5, line 186).
